# A Method for Assessing Flood Vulnerability Based on Vulnerability Curves and Online Data of Residential Buildings—A Case Study of Shanghai

**Zhuoxun Li** [1], **Liangxu Wang** [1,*], **Ju Shen** [2], **Qiang Ma** [3] and **Shiqiang Du** [1,*]

1   School of Environmental and Geographical Sciences, Shanghai Normal University, Shanghai 200234, China
2   Shanghai Emergency Management and Chemicals Registration Center, Shanghai 200020, China
3   Resource and Environment College, Anhui Science and Technology University, Chuzhou 233100, China
*   Correspondence: wangliangxu@lzb.ac.cn (L.W.); shiqiangdu@shnu.edu.cn (S.D.)

**Abstract:** Flood vulnerability is the key to understanding and assessing flood risk. However, analyzing flood vulnerability requires sophisticated data, which is usually not available in reality. With the widespread use of big data in cities today, it is possible to quickly obtain building parameters in cities on a large scale, thus offering the possibility to study the risk flooding poses to urban buildings. To fill this research gap, taking Shanghai as an example, this study developed a new research framework to assess urban vulnerability based on vulnerability curves and online data of residential buildings. First, detailed information about residential buildings was prepared via web crawlers. Second, the cleaned residential building information fed a support vector machine (SVM) algorithm to classify the buildings into four flood vulnerability levels that represented the vulnerability curves of the four building types. Third, the buildings of different levels were given vulnerability scores by accumulating the depth–damage ratios across the possible range of flood depth. Further, combined with the unit price of houses, flood risk was assessed for residential buildings. The results showed that the F1-score for the classification of buildings was about 80%. The flood vulnerability scores were higher in both the urban center and the surrounding areas and lower between them. Since 1990, the majority of residential buildings in Shanghai have switched from masonry–concrete structures to steel–concrete structures, greatly reducing the vulnerability to floods. The risk assessment showed decreasing risk trend from the center outward, with the highest risk at the junction of the Huangpu, Jing'an and Xuhui districts. Therefore, this framework can not only identify the flood vulnerability patterns but also provide a clue for revealing the flood risk of residential buildings. With real estate data becoming increasingly accessible, this method can be widely applied to other cities to facilitate flood vulnerability and risk assessment.

**Keywords:** building structure; vulnerability curve; support vector machines; flood risk; damage assessment

## 1. Introduction

China is one of the countries suffering the most serious natural disasters in the world [1,2]. Among them, flood and waterlogging disasters occur frequently, affecting a wide range of the population and causing heavy losses [3]. Moreover, flood risk is likely to increase owing to a combination of climate change, sea level rise (SLR) and land subsidence [4,5]. Flood risk and its likely increasing trend are particularly acute in urban areas, where a large number of people and wealth are accumulated [6–8]. On the other hand, urbanization is accelerating, with the world's urban population expected to increase from 3.6 billion in 2011 to 6.7 billion in 2050, which is likely to exacerbate the potential impact of flooding on cities [9]. Therefore, reducing flood disaster losses is a key to promoting sustainable development and requires a thorough understanding of flood risk [10–12].

Flood risk has three core components, namely, hazards (and inundations), exposure and vulnerability [13–15]. For flood hazards, hydrodynamic models were developed [16–18] to simulate the inundation processes and identify flood-prone areas [19]. However, the spatial variability of flood-bearing elements caused spatial heterogeneity, local correlation and neighborhood non-stationarity to appear in the risk [20]. Thus, it is challenging to scientifically and rationally measure flood risk. Extensive studies on flood exposure have also been conducted in recent years based on remote sensing data and radar image data [21–24]. In comparison, the research on flood vulnerability remains relatively limited.

Flood vulnerability describes the potential damage degrees given a magnitude of flood hazard, which is influenced by the characteristics of the exposed assets [25,26]. Regarding residential buildings, the material, structure, height, construction time and location affect the level of flood vulnerability, which determines the loss degrees of residential buildings suffering from flooding [27]. Moreover, flood vulnerability changes with the evolution of residential buildings in a rapid urbanization process [28].

In general, the methods of vulnerability assessment include vulnerability matrices [29,30], vulnerability indicators [31] and vulnerability curves [32]. Although the former two methods are also widely used in assessing the vulnerability of residential buildings, they are subjective and sensitive to the understanding of researchers [33,34]. In comparison, the vulnerability curve describes the different loss ratios as a function of flood depths and the exposed residential buildings, which is more objective and quantitative [35,36]. Given a particular residential building, it is generally expressed as a series of damage ratios ranging from 0 (no damage) under low flood depth to 1 (total loss) under extreme high flood depth [37,38]. Historical disaster data can be collected in a variety of ways, such as questionnaires [39], to establish the vulnerability curves [40,41]. Residential buildings with different characteristics, e.g., materials, structures and construction time, should have varied vulnerability curves [42,43]. However, a vulnerability curve cannot give a general description of the flood vulnerability for different residential buildings, as it represents a series of values along changing flood depths.

On the other hand, big data, which has emerged in recent years, can provide a feasible tool to acquire building characteristics. Englhardt et al. developed an approach for large-scale flood vulnerability assessments using object-based information about buildings and the built environment based on the fine-scale ImageCat dataset [44]. Although the ImageCat dataset is not openly accessible, this study provided an important case for using big data to enhance the description of flood vulnerability. Similar cases also included the use of news media photographs at the time of a disaster to detect the entire extent of the flooded buildings [45]. The combination of structured, semi-structured and unstructured data from online big data can help to mine information about the residential buildings that determine the flood vulnerability, such as materials and structures [46,47]. This new tool can also extract building profiles and values based on online real estate big data [48]. However, there is a lack of adequate research on how to explore this type of data for flood vulnerability and risk assessment.

To fill the research gap, this study proposed a new research framework to extract building characteristics from online big data of real estate and calculate vulnerability scores for residential buildings by combining vulnerability curves for the corresponding building types. The result was then combined with exposure component to assess the flood risk. We chose Shanghai as a case to apply this framework for assessing the vulnerability and flood risk of residential buildings.

## 2. Materials and Methods

### 2.1. Study Area

Shanghai, with a terrestrial area of 6340 km$^2$ (Figure 1a), is one of the megacities that are vulnerable to extreme rainstorms and storm surges [49,50]. It is located on the route of western Pacific typhoons, which cause strong winds, high tides and heavy rains. On the other hand, it is one of the world's financial centers, home to a large population and mega

economy. In 2021, it had a population of 24.9 million, accounting for 1.76% of the total Chinese population while producing 3.78% of the national GDP. In the past few decades, the ratio of the urban population to the total has increased from 59% in 1978 to 89.3% in 2021 [51]. The central urban area is delineated by the Outer Ring Road (Figure 1b). It was selected as the study area for two major reasons. First, it has the longest development history and is also undergoing extensive urban renewal. The percentage of the impervious surface in this area is approximately 81% [52]. Second, a long history of land subsidence and river disappearance has rendered this low-lying area prone to flooding [53].

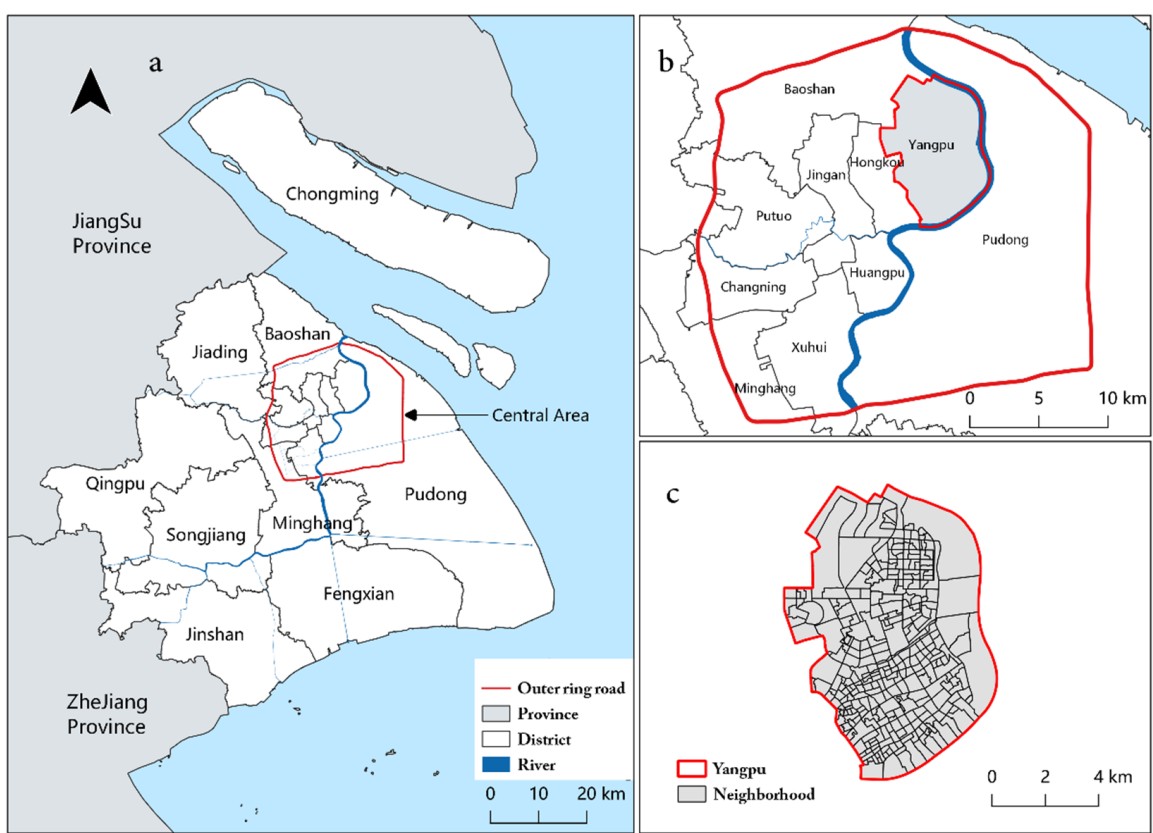

**Figure 1.** Geographical setting of Shanghai: (**a**) administration levels; (**b**) inner center; (**c**) neighborhood levels in the district.

*2.2. Data*

Three major datasets were used in this study. First, the characteristics of residential buildings were obtained through the web crawler, namely, the community information provided by the real estate firm Home Link China (https://sh.lianjia.com/ (accessed on 26 December 2019)). Their services cover many cities in China, making the methodology widely applicable. Multiple attributes can be acquired, including the geographic location, construction time, story and structure type. The dataset covered a total of 21,480 communities in Shanghai. After cleaning and consolidation, the total number of communities with complete information was 10,772, accounting for 50.14% of the city. In the central area, there were 7166 in the cleaned dataset which covered 84% (2317) of the 2743 neighborhood committees (Figure 2). Therefore, the central area was chosen as the study area for further vulnerability and risk assessment.

Second, we employed a 100-year pluvial flood hazard map of Shanghai, which was produced using the hydraulic FloodMap 2D model [54]. The flood depth map had a horizontal resolution of 5 m and the maximum depth was 6.82 m.

Third, we chose the 2019 LandScan population density dataset, which was produced by the Oak Ridge National Laboratory using a linear regression model with a spatial resolution of approximately 1 km$^2$.

These two data sets were used for correlation analysis with the final calculated flood risk to explore the applicability of the methods in this study.

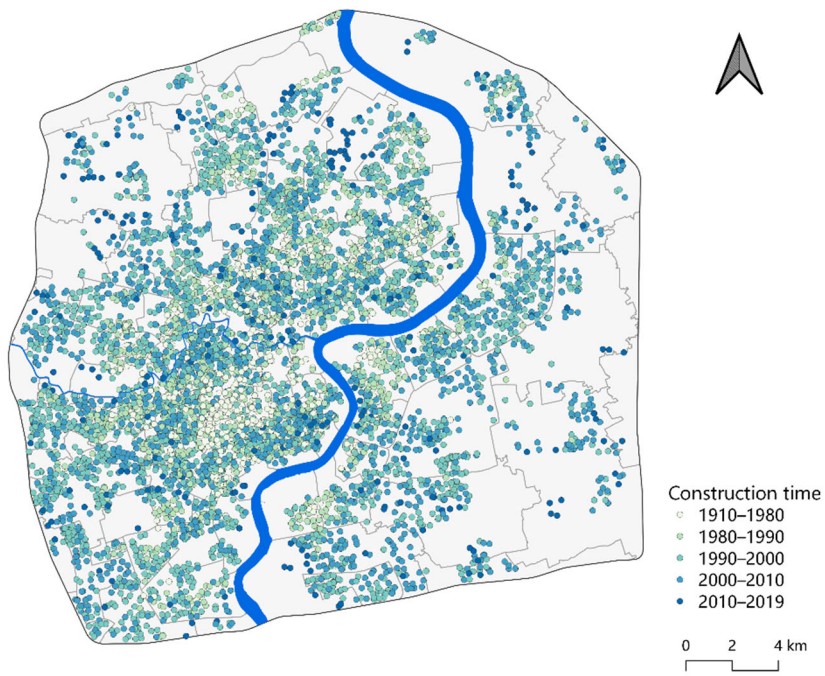

**Figure 2.** Distribution of community locations by years.

*2.3. Methods*

This study aimed to develop a new framework to assess the urban flood vulnerability and, in turn, the flood risk using online data of residential buildings, which could overcome the data scarcity problem and quickly acquire fine-scale vulnerability information about residential buildings to support flood risk assessment. Figure 3 shows the research framework. The main steps were as follows. First, the location and attributes of the residential buildings were extracted through the web crawler from the real estate website (https://sh.lianjia.com/ (accessed on 26 December 2019)), which was followed by data cleaning. Second, a series of vulnerability curves were employed to calculate a comprehensive vulnerability score to represent the buildings' ability to withstand disasters. Third, the residential buildings were classified into different types, matching the vulnerability curves based on the information from the first step using machine learning models. Fourth, flood risk was assessed by combining the vulnerability results, flood map and information on housing prices. Finally, the fine-scale spatial patterns were identified from the flood vulnerability and risk results.

2.3.1. Extracting Information about Residential Buildings

First, we evaluated the quantities of the above data and extracted the types of data features after we found the online housing data platform. Second, we analyzed the laws of the frame and URL construction of the web page, found the location of the housing features in the web page elements and then performed web page parsing. Finally, after crawling all the data, the data was cleaned to obtain the information of all the neighborhood committees with complete building characteristics.

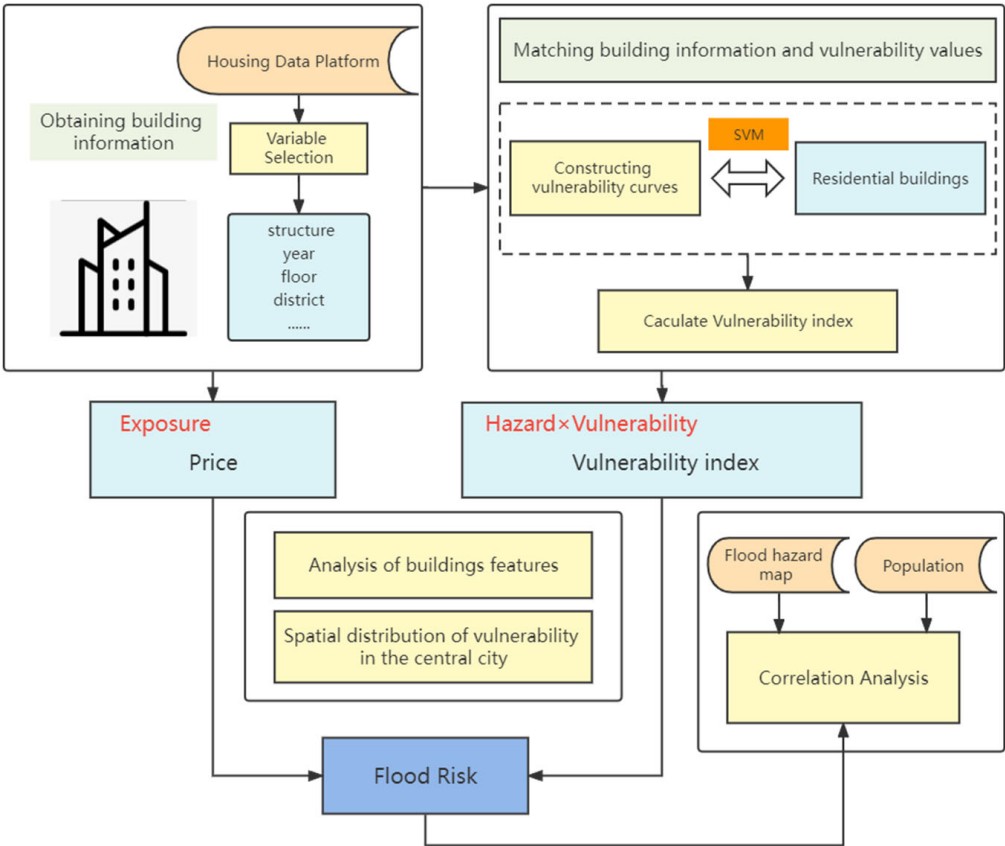

**Figure 3.** The research framework of flood vulnerability using housing trading platform big data.

### 2.3.2. A Score Integrating the Vulnerability Curves

Based on the vulnerability curves, we proposed a new method to study the vulnerability data. The vulnerability curve was used to describe the loss rate of the disaster body under potential water depths (Figure 4). Therefore, it was not a specific value but had different values across different flood depths. It could not directly present the vulnerability level of an element, although the series of damage ratios would be typically higher for a vulnerable element than for a strong element. In order to study disaster vulnerability in a manner that only relied on the characteristics of the exposed element and was independent of the flood depth, we used the definite integral of the vulnerability curve for a building across different flood depths to express its vulnerability. Therefore, it synthesized the loss ratios of a vulnerability curve into a specific value:

$$Vulnerability = F(x) = \int_0^{\max} f(x)dx \tag{1}$$

where $V$ is the vulnerability curve that connects the flood depth and loss ratios (percentage given a depth) through the integral function $F(x)$, $x$ refers to the variable of potential flood depths, $f(x)$ represents the loss ratios of the residential building given a flood depth $x$ and $dx$ is the integrand function. Generally, the loss ratio could reach a maximum of one with a maximum flood depth [55].

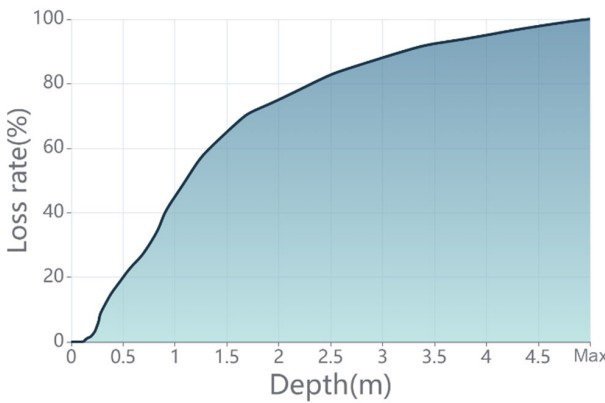

**Figure 4.** Stage–damage curves for building-material-based vulnerability.

### 2.3.3. Classifying Residential Buildings Based on Their Vulnerability Information

Following Englhardt et al. [44], the residential buildings could be divided into four categories of flood vulnerability mainly based on their building materials, i.e., non-engineering buildings or informal buildings, wooden buildings, masonry–concrete structures and steel–concrete structures. Furthermore, following flood vulnerability studies in Shanghai [56], the buildings were classified into four categories: bungalows (Level 1), multistorey houses (Level 2), high-rise houses (Level 3) and mansions (Level 4). In Shanghai, urban development has its own characteristics. The city had a mix of modern and aged buildings, which caused different flood vulnerability. Therefore, the construction time, number of floors and building area were also considered, in addition to the building materials, to classify the buildings into the four categories of flood vulnerability.

The building characteristics were then fed into machine learning for classification. In this study, 700 pieces of data were randomly selected for manual annotation through the characteristics of the district, construction time, storey and building structure. We also combined the real pictures taken on the platform (https://sh.lianjia.com/ (accessed on 26 December 2019)) as an additional factor for the judgment. As these features were not collinear, they were trained directly using three machine learning methods of random forest (RF), support vector machine (SVM) and extreme gradient boosting (XGBoost). After comparing the F1-scores of the three methods, SVM was selected as the best classification method.

SVM is a machine learning method that was proposed by Corinna et al. [57] and is based on the Vapnik–Chervonenkis dimension (VC dimension) theory of statistical learning theory and the structural risk minimization principle and seeks the best compromise between model complexity and learning ability to overcome limited sample information. An SVM classifier maps the original nonlinear data to a high-dimensional space, finds a global optimal classification hyperplane that meets the classification requirements and uses the optimal classification hyperplane to distinguish the vulnerability levels of several types of buildings to ensure the maximum classification interval. We used the linear kernel function to train the classification model. The F1-score for the training set was about 80%. The classification results are shown in Table 1.

**Table 1.** Classification description of architectural features for four types of vulnerability curves.

| Level | Description | Number |
|---|---|---|
| Level 1 | Using brick and wood, with a few stories and a building age of more than 50 years | 4678 |
| Level 2 | Using brick and concrete or reinforced concrete, a few stories and slightly older | 1657 |
| Level 3 | Using reinforced concrete, with multi stories and aged around 20 years | 2721 |
| Level 4 | Using reinforced concrete, with multi stories, spacious area and built in the 21st century | 2415 |

2.3.4. Flood Risk Assessment

Disaster losses are dependent on a combination of vulnerability and economic elements within the region. For residential buildings, the most direct element that reflects the economic value is the house price, which is also acquired from the online data on real estate. In this study, we used the house prices of exposed residential buildings and the derived vulnerability to determine the flood risk. The formula is as follows:

$$Risk = Flood\ hazard \times Exposure \times Vulnerability \tag{2}$$

where *Risk* is the flood risk in terms of the potential economic damage (CNY/m$^2$), *Flood hazard* refers to the inundation depth; *Exposure* refers to the unit price of exposed residential buildings and *Vulnerability* refers the result of Equation (1).

2.3.5. Spatial Pattern Identification

The local Gi* statistic that was proposed by Ord and Getis [58] was used to identify hot spots (high values) and cold spots (low values) with statistical significance. It not only considers the number of elements, spatial location and adjacent elements but also considers the data attributes and weights of spatial units. At present, it is widely used in the research of economic geography, traffic accident analysis, population distribution, urban development, the spatial layout of retail formats and other fields. It can be calculated as follows:

$$Gi^* = \frac{\sum_j^n w_{ij}(d)x_j}{\sum_{j=1}^n x_j} \tag{3}$$

where $w_{ij}$ is an *n*-by-*n* symmetric matrix with 0 or 1 values and represents the spatial relationship between neighborhood *i* and the other neighborhoods; a value of 1 indicates that neighborhood *j* is a neighbor to neighborhood *i*, while 0 means that *j* does not neighbor *i*, which is determined by the distance *d* between them. $x_j$ refer to the values of neighborhood *j*.

**3. Results**

*3.1. The Structural Characteristics and Changes in the Vulnerability of Urban Buildings*

According to the building structure types collected by the platform, there were seven main types of structures. As shown in Figure 4, steel–concrete structures and masonry–concrete structures were mainly used, while some buildings were classified as other structures, such as masonry–timber structures, frame structures and steel structures, which accounted for less than 1% of buildings. This disparity reflects the progress of urbanization in Shanghai. Nearly 60% of the residential areas in the central urban area were solid steel–concrete structures, while some old-fashioned houses were masonry–concrete structures, which were found in some old-fashioned residential areas left during the development of Shanghai. Taking Xuhui District as an example (Figure 5), the numbers of residential areas with masonry–concrete structures and steel–concrete structures accounted for about 50% respectively. However, the proportion was different in the sub-districts. The Lingyun Road Sub-district mainly consisted of masonry–concrete structures, while most buildings in the Xujiahui Sub-district had a steel–concrete structure.

Regarding the time scale, the obtained data also ranged from 1910 to 2019. The main construction time was concentrated after the 1980s while the earliest construction was built in 1910 (Figure 6). In view of the urban development process of Shanghai, four cut-off points of 1980, 1990, 2000 and 2010 were selected to divide the construction time into five periods (Figure 7). In the two stages of 1910–1980 and 1980–1990, the city mainly contained masonry–concrete structures. After the 1990s, the urban building structure began to change. Steel–concrete structures gradually replaced the masonry–concrete structures and their proportions were basically the same. After 2000, there was only a small percentage of masonry–concrete structures. The city mainly focused on the construction of residential buildings with steel–concrete structures.

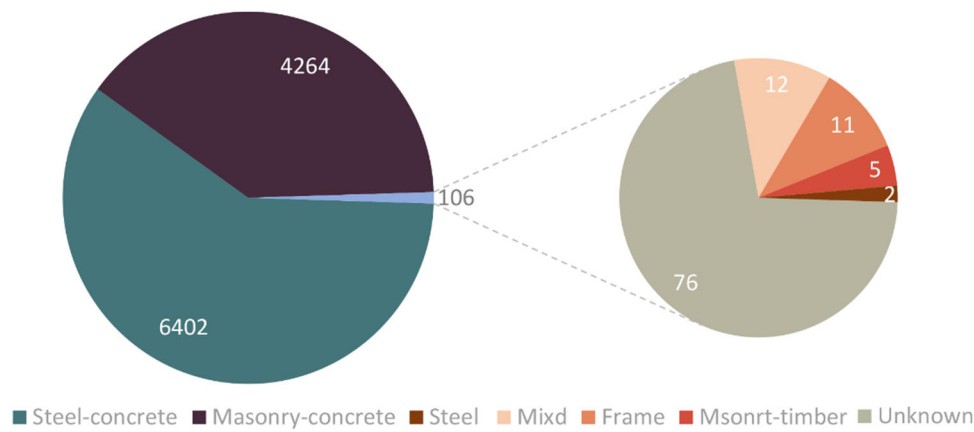

**Figure 5.** Number of different building structures.

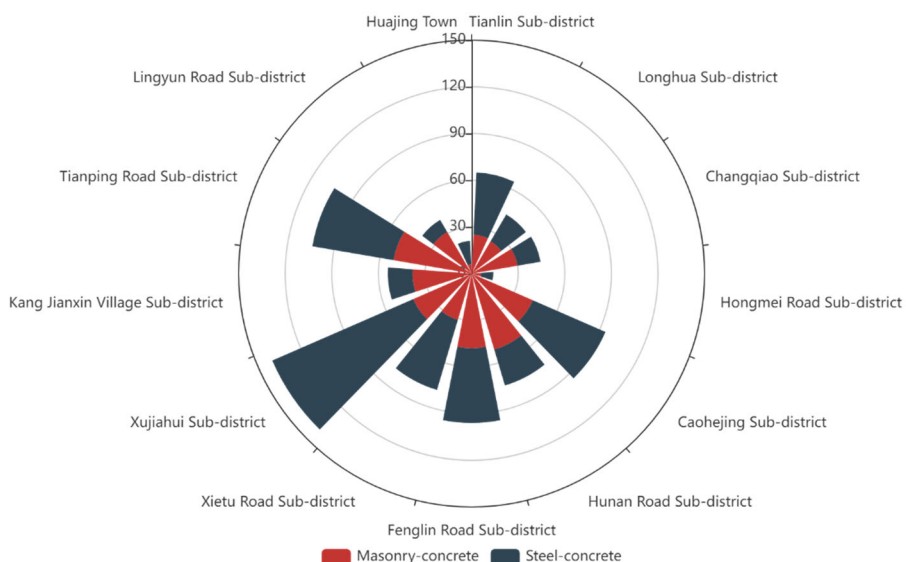

**Figure 6.** Structure type distribution in the sub-districts of Xuhui District.

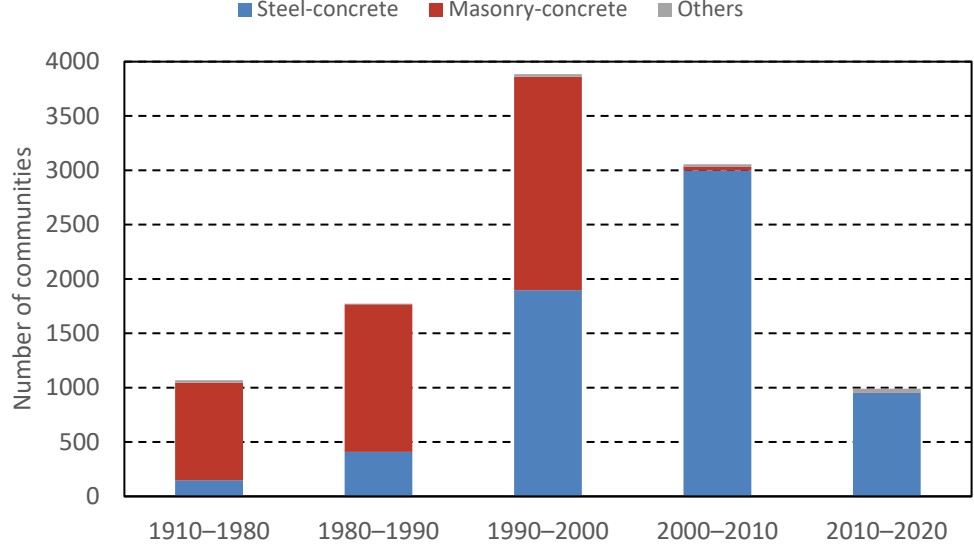

**Figure 7.** Quantities and proportions of communities in different periods.

*3.2. Spatial Distribution of Regional Vulnerability in Central Shanghai*

We used the ring road outside the central urban area as the boundary to analyze the distribution of the vulnerability index at the sub-district and neighborhood committee scales. The central urban area was an area with a weak ability to deal with natural disasters, such as rainstorms and floods, which was the target of the research.

By dividing the vulnerability index into five levels according to the natural breakpoint classification (Figure 8), the vulnerability index was classified into low, medium-low, medium, medium-high and high vulnerability zones. The vulnerability distribution along the Huangpu River is obviously high in the west and low in the east. A large number of medium-high and high vulnerability areas existed west of the Huangpu River (Puxi). The three highest areas were Yangpu Yanji Xincun Street (Figure 8(b①)), Xuhui Huanan Road Street and Huangpu Ruijin Second Street (Figure 8(b②)). Meanwhile, the higher vulnerability areas were clustered into several areas, distributed in the northeast corner of the inner ring and the old city of Shanghai south of Suzhou Creek. The northwest corner of the central city was the only area with low vulnerability in Puxi. Compared with Puxi, Pudong had lower vulnerability.

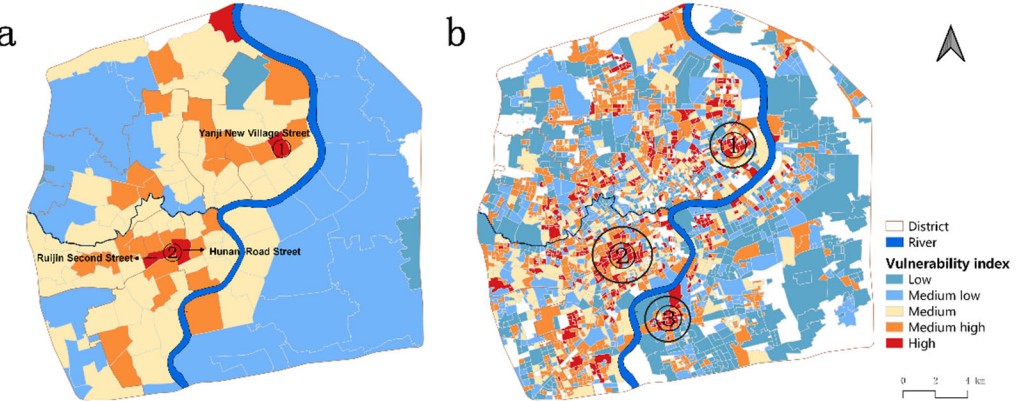

**Figure 8.** Distribution of vulnerability indexes at the (**a**) sub-district and (**b**) neighborhood committee scales.

At the neighborhood committee scale, more detailed observations were made of the heterogeneity of the vulnerability in the central area, especially in the large sub-districts. Medium-high and high vulnerability zones were more widely distributed and existed in almost every region. Moreover, Zhou Jiadu Street and Shanggang Xincun Street in Pudong had distinct clusters of high vulnerability zones (Figure 8(b③)).

We further calculated the proportions of vulnerable areas at different levels for both scales (Figure 9). We found that the proportion of low and medium-low vulnerable areas has not changed overall, but the proportion of medium vulnerable areas had decreased by more than half, while the proportion of medium-high and high vulnerable areas had increased significantly.

The local Gi* statistics identified the spatial agglomeration of vulnerability (Figure 10). Hot spots and cold spots accounted for 877 (31.97%) and 505 (18.41%) of the neighborhood committees, respectively. The hot spot area extended from the middle of Yangpu to the southern part of Jing'an, south of Suzhou Creek, the northern part of Xuhui and the western part of Huangpu along the inner ring. A major cold spot area was at the junction of Huangpu River and Suzhou Creek in the central area, extending from the North Bund to the Bund. It used to be the main part of Shanghai's urban built-up area in the past, but it became a commercial center with only a small number of residential functions at present.

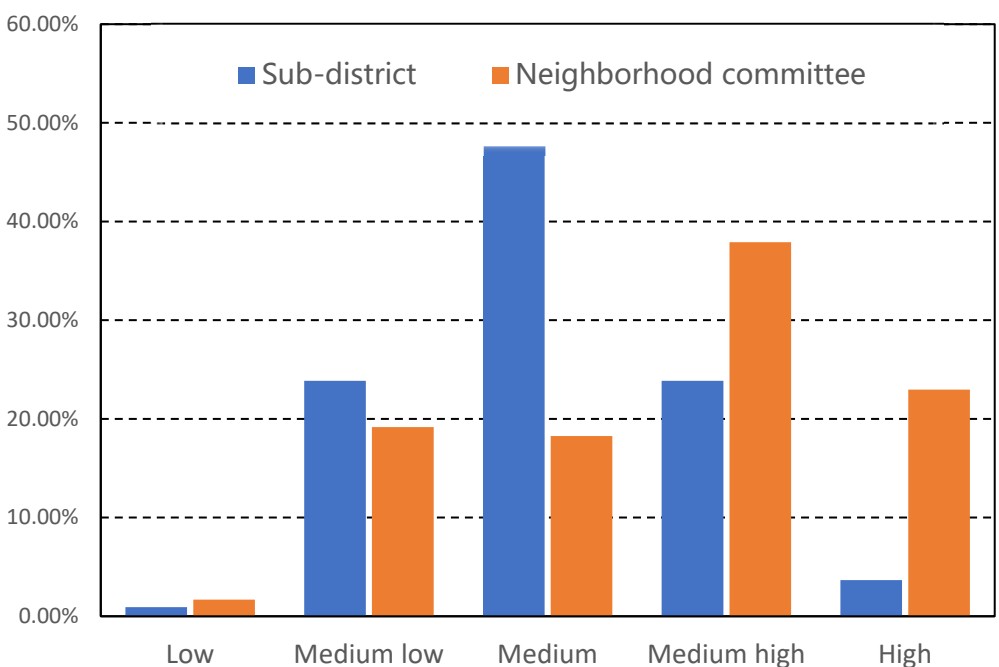

**Figure 9.** Histogram of sub-districts and neighborhood committees across different vulnerability levels.

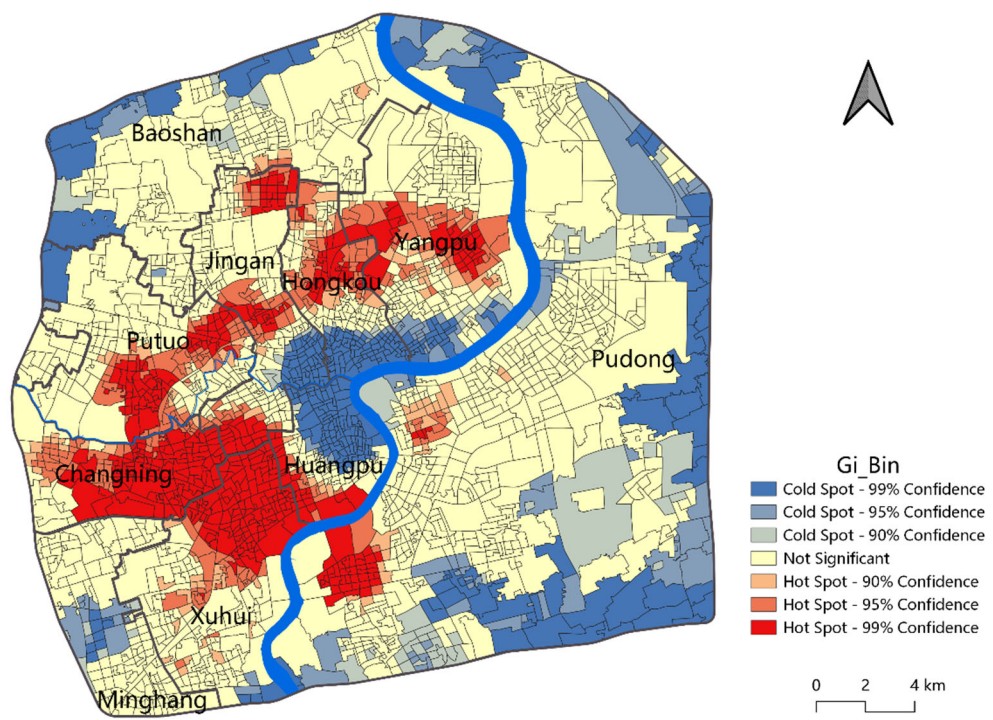

**Figure 10.** The hot/cold spots of flood vulnerability in central urban areas.

### 3.3. Flood Risk Assessment

Figure 11 shows the combination of the vulnerability index and house unit prices in the central city, with low vulnerability and high prices accounting for about 21.7% of the total, while high vulnerability and high house prices were 41.4% of the total. The difference of nearly twice as much illustrated that for city center areas, the level of risk could not be what affected the value of the location. The most hazardous combination of high vulnerability and high prices was mainly located in the core area of Puxi. These areas should be priority areas for the agendas addressing the mitigation of flood hazards.

The southwestern parts of the center city, including Xuhui, Huangpu and Changning, had medium or higher levels for both prices and vulnerability. In comparison, Yangpu and Hongkou were less expensive, though they had the same higher vulnerability.

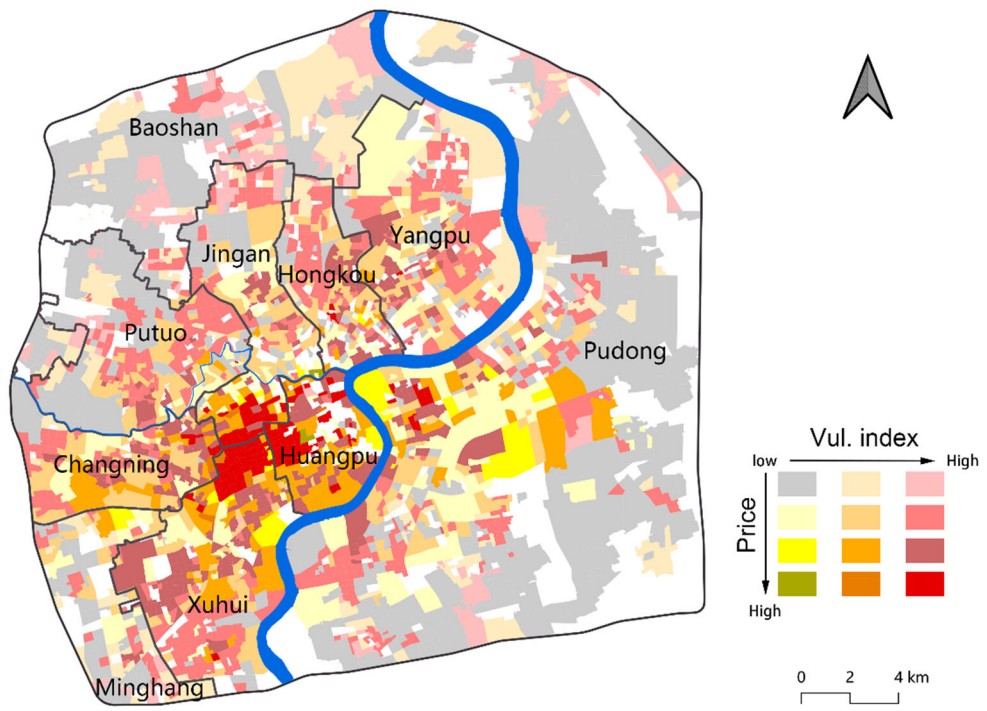

**Figure 11.** Bivariate depiction of the vulnerability index and house unit price.

Figure 12 shows the flood risk at the neighborhood scale; it was found that the results were slightly less heterogeneous in space. In general, it showed a characteristic of spreading from the center to the periphery. The risk in the lower reaches of Huangpu River was also significantly higher than that in the upper reaches. It was predicted that the damage will be higher in the Puxi core when a disaster strikes due to the denser and more complex built environment with more complex properties and correspondingly higher values.

To further analyze the applicability of the result obtained from the research framework of this study, we introduced population data and waterlogging data to analyze their correlation. After verifying that the data were normally distributed, their correlations were analyzed using Pearson correlation coefficients, and the results passed the 1% significance test (Table 2).

**Table 2.** Correlation between the flood risk and the simulated depth of water accumulation and population density.

| | Simulated Depth | | Population Density | |
|---|---|---|---|---|
| | **Street** | **Neighborhood Committee** | **Street** | **Neighborhood Committee** |
| Flood Risk | 0.515 | 0.226 | 0.644 | 0.647 |

It was found that the depth of ponding was moderately correlated with disaster losses at the street and town scale, while only weakly correlated at the neighborhood scale. This reflected the result that the areas that were more affected by flooding were also high-loss areas.

Areas with high population density were exposed to higher total risk and expected to incur higher disaster losses when hit by floods. The correlation between regional population density and disaster loss values at both the street and town and neighborhood committee

scales exceeded 0.6, showing a strong correlation, and the spatial distribution of population density and the spatial variation of disaster loss were generally consistent.

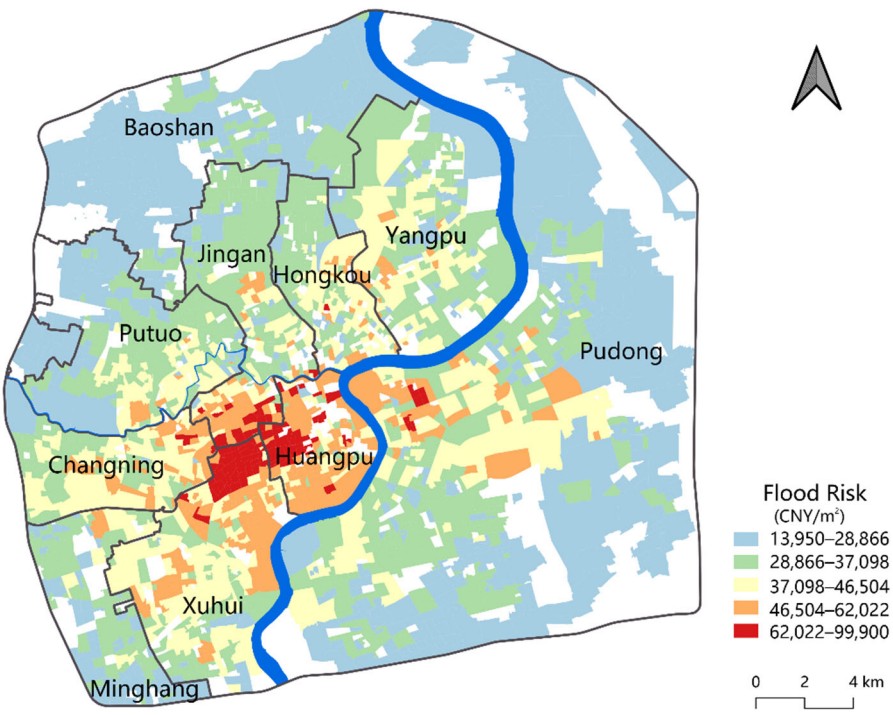

**Figure 12.** Flood risk at the neighborhood committee scale.

## 4. Discussion

### 4.1. Flood Vulnerability Reduction and Consistency of Urban Planning

The frequency of fluvial and coastal storm flooding has decreased greatly, while urban pluvial flooding produced by rainstorms has increased dramatically in Shanghai in recent decades [59]. This is not a singular case. With climate change, dangerous disasters have appeared all over the world, which makes disaster reduction a topic of worldwide interest [60].

Facing flood risk in different regions, various methods of assessment have evolved, from basic single-parameter to multiparameter approaches, such as multivariate or indicator-based models [61]. However, many studies need to collect corresponding data and build models according to the local environment. Therefore, it is necessary to construct a method to evaluate the overall risk using fewer data to consider the universality of the method, where the vulnerability index was selected as the effective index.

In this work, we first collected information about residential buildings in Shanghai. The result showed that the buildings were mainly composed of masonry–concrete structures and steel–concrete structures. What is more, there was an obvious transition trend from the former to the latter over time, which was mainly concentrated in the stage during 1990–2000. This was also the time of Shanghai Pudong Development and urban planning for the new century. At that time, taking the development and construction across the Huangpu River in the Pudong area as the symbolic feature, large-scale land leasing and reconstruction of dangerous sheds and simple houses led to the construction of many new residential buildings and promoted the evolution of the urban form. The transition of building structures significantly improved the protection ability against flood disasters. This was also reflected in the time change of the vulnerability. At this stage, the vulnerability index decreased by 8.95%.

In the central city, the streets with high vulnerability in Puxi were mostly mature residential areas dominated by old buildings with masonry–concrete structures and six floors or less. Streets with low vulnerability were mainly concentrated in areas with some new residential buildings. The area along the Huangpu River with a low vulnerability

index mainly involved residential demolition and renewal. Today, it is dominated by commercial and tourist functions.

Most sub-districts in the Pudong New District were medium-low vulnerable zones because the area in the outer ring was the best-developed region in Pudong New District. Four functional blocks were established here. During the rapid development and expansion of Shanghai, a large number of new houses were established here. However, at the scale of the neighborhood committee, there was also a block with a highly vulnerable area in it, which was located on Shanggang Xincun Street and Zhoujiadu Street. This was also reflected in Figure 11, with an obvious hot spot area. This was because there was a wave of house relocation during the 2010 Shanghai World Expo. This was originally a rural area and basically became a residential area as early as the 1990s. After the World Expo, this area was also reserved as a future area to be developed. It will be a new growth site for Shanghai's development in the next 50 years. At the same time, it will also be a practice area and a leading area for Shanghai's economic development and transformation. Hence this area is still undeveloped for the sustainable development of Shanghai over the next few decades.

### 4.2. Big Data Offers New Opportunities for Disaster Risk Research

Vulnerability is the core to understanding natural disaster risk; however, it is also a vague concept regarding the evaluation indicators. The mainstream research method in the past used multiple indicators to construct an evaluation system, which could analyze the vulnerability magnitude of a region. Nevertheless, a unified indicator system has not yet appeared. In addition, too many indicators also cause difficulties in collecting data for inter-regional comparison [62]. However, the advent of the big data era allows us to use massive amounts of data combined with expertise to monitor potential vulnerability in data-deficient areas and to assess disaster losses in cities [63,64].

In this study, we used the unique data source of building data from the online housing trading platform to analyze the vulnerability and identify the spatial pattern underlying the vulnerability within a city. This method reduced the reliance on extra data as much as possible. In particular, it can also be applied to fine-scale administrative divisions, which are typically lacking in sufficient data for indicator-based vulnerability analysis.

Based on this, we proposed a new research framework that can extract the information of at-risk elements using current online big data and then estimate the vulnerability of each research unit using a machine learning model. This framework is useful for building fine-grained disaster studies at larger spatial scales. At present, the online housing data has a certain coverage in China, and it is also applicable to the first-tier cities that have real estate agency services. In addition, as data are updated and supplemented, a basic information database based on housing data can be established [65]. Combined with disaster data and other deep learning methods, such as neural networks, the evolution trend and influencing factors of urban vulnerability can be analyzed. This can help to develop reasonable planning, observe the evolution pattern of vulnerability, and identify vulnerable links and their geographical distribution, thus playing an important role in urban and disaster emergency management.

### 4.3. Limitations and Future Perspectives

This study had some insufficiencies that need further investigation. The first problem was data completeness because the complete building data only accounted for about 60% of the total for the entire Shanghai city. In terms of building types, the data used here only included urban dwellings, not other building types, while in the suburbs, some houses were non-tradable, and thus, there was less trading there, meaning that the platform lacks building information for the suburbs. As a compromise, we used the central area instead of Shanghai as the study area since the acquired data could cover 84% of central Shanghai. We need more comprehensive and detailed building data to refine the characteristics of residential buildings throughout the city. In the future, using the results

of the country's first national comprehensive natural disaster risk survey as a data source, more data could be available for assessing vulnerability. However, we believe that the methodology we developed in this study could play a key role, as the source data could be updated consistently.

Second, in the current study, the smallest unit was the community, which has not reached the granularity of a single disaster. If the attributes of individual buildings are used, this will reveal regional vulnerability differences more accurately. This study assumed that the property information of housing reflected the vulnerability of the whole region, and the value of its vulnerability index was fixed. The risk analysis of the first floor of a building is also an extraordinarily important element regarding flooding. At present, due to data limitations, we are currently unable analyze them separately either.

Finally, from the perspective of the generalization of the study, on the basis of the existing data, only a case study of Shanghai was undertaken. When calculating the vulnerability and loss of other cities in the future, we should consider the universality of the vulnerability curve and compare it with the historical loss.

## 5. Conclusions

In this study, we developed a research framework to use the housing data from online trading platform for evaluating flood vulnerability, taking Shanghai as a study case. The framework obtained the residential building attributes through a web crawler. Then, it analyzed the structural characteristics, used machine learning methods to combine with the established vulnerability curve correspondence to obtain the vulnerability index, analyzed the vulnerability distribution characteristics at different spatial scales and finally calculated the flood risk. The results showed that, in general, the vulnerability varied significantly with the change in residential building materials. The vulnerability in the city center, which was associated with high house unit prices, indicated the highest flood risk. Spatially, the flood vulnerability scores were higher in both the urban center and the surrounding areas, and lower in between. The overall vulnerability of Puxi was higher than that of Pudong. Temporally, the vulnerability scores in Shanghai declined with the residential building majority changing from masonry–concrete structure to steel-concrete structure. The high heterogeneity of the vulnerability of the central city and the reduction in the overall score reflected the evolution of the city from a staggered distribution of old and new residential buildings to a gradual renewal process during the development process. The risk assessment showed a decreasing trend from the center outward, with the highest risk at the junction of the Huangpu, Jing'an and Xuhui districts. The flood risk was developed for different vulnerability indexes, which enabled us to estimate the flood damage for each house type as a monetary value. This allowed for evaluation of the adaptation options of each house type to reduce the disaster risk and can help to better manage flood risk in the future.

The differences in vulnerability are thought to be related to the urban development process and probably also exist in other cities in China and worldwide. Our research can be used as critical information to better identify communities that may be disproportionately affected and in need of additional assistance during extreme floods, thereby helping the government to monitor risk areas and plan for the preparation and management of disaster relief resources. In the future, with census data, such as the First Comprehensive Survey on Natural Disaster Risks in Shanghai, becoming publicly accessible, vulnerability classification will be detailed.

**Author Contributions:** Conceptualization, S.D. and L.W.; methodology and software, Z.L., S.D. and L.W.; validation, J.S. and Q.M.; formal analysis, investigation and data curation, Z.L. and J.S.; resources, supervision, project administration and funding acquisition, S.D. and L.W.; visualization and writing—original draft preparation, Z.L.; writing—review and editing, S.D. and L.W. All authors have read and agreed to the published version of the manuscript.

**Funding:** This research was funded by the National Natural Science Foundation of China (grant nos. 41871200 and 41730642) and the National Key Research and Development Program of China (2017YFC1503001).

**Institutional Review Board Statement:** Not applicable.

**Informed Consent Statement:** Not applicable.

**Data Availability Statement:** Not applicable.

**Conflicts of Interest:** The authors declare no conflict of interest.

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
