# Peer review of "A Method for Assessing Flood Vulnerability Based on Vulnerability Curves and Online Data of Residential Buildings—A Case Study of Shanghai"

_water, doi:10.3390/w14182840_

Round 1

Reviewer 1 Report

The authors presented a vulnerability assessment of Shanghai. Although they claim that the research is about "big data" analysis, no information is given about the size of the dataset. Most likely, the data size in the study wouldn't exceed a couple hundred gigabytes, which could definitely be a big data analysis in the early 2000s. Currently, tera or petabytes of data are considered big data. So, I would revise the title and remove the emphasis on big data.

There are critical points that need clarification. In particular, data sources and vulnerability methodology require additional explanations. See my comments below.

Line 95, Authors should clearly state the data source and the metadata. Python is obviously a tool to extract the data, but the procedure should be clearly explained. Is there any data API available to the general public? It seems the website doesn't provide an API service. I found a Python repository called "LianJiaSpider" that can collect property information, but the release date is 2018, which does not coincide with the publication date of this manuscript. In line 122, The Web crawler seems to be an integral part of the vulnerability assessment framework, but the given information is very limited. Describe what it is and how it works. Is there any public GitHub repo?

Line 141, the developed vulnerability function, is a bit interesting. In section 2.3.2., the authors mentioned depth-damage functions and explained how they came up with a new method that again relies on the depth and the function of fit. Authors must elaborate on why their method is a new approach. The function is not well described. What is f(x)? Function of fit? Not clear at all. 

Line 145, section 2.3.3, doesn't give much information about how building materials are involved in vulnerabilities. It's clear that the effects of floods on properties depend on how they were built, but the article doesn't give any weights. Weights should have been emphasized why building material type matters when considering flood vulnerability.

Line 244, section 3.2, doesn't provide clear information on the curve index and rate of change in Figure 8. The Y-axis has no title for understanding the figure. Elaborate.

In Figure 13, the authors presented a quantified flood risk in the community. Risk is generally estimated based on the probability of scenarios. There are no probabilistic flood maps that have been used in the study. The authors failed to explain 1) how quantified losses are estimated? 2) How the probability is involved in the estimation?

How first floors are evaluated for vulnerability? In Shanghai, there are many stores located on the first floor of buildings. How they were filtered to analyze only residential buildings?

Reviewer 2 Report

The authors present a very interesting topic of assessing flood vulnerability of residential buildings using big data. The paper was very promising, however, there are several flaws. First and most important there are several points in the methods section that are poorly explained or not explained at all.   For example: How did the authors measure the vulnerability? What data of losses do they use to link the building characteristics with the losses?

 The authors claim that they assess the final disaster risk (line 126), but they do not mention what data they used to evaluate the hazard and the exposure, component of risk as they mention in lines 44-45.

 The authors fail to explain how they integrate the building characteristics in the vulnerability index since in the equation they present in line 141 the vulnerability only depends on the flooding depth. In line 160 they said that “the curves were integrated to obtain their combined vulnerability values”, I assume integrating the building characteristics to the integral function using flood depth, but they do not mention how.

 In section 2.3.4 Building classification model authors need to justify why they use those building characteristics to make the classification. They mention that they combine the real pictures as an additional basis for judgment (lines 169-170), did they review all, or how did they decide which one to review?

They mention that they compared three classification methods: RF, SVM, and XGBoost, but do not explain how and why they selected SV.   

 In section 2.3.5 Flood risk assessment, authors now give a different equation to calculate risk, the use of exposure, and vulnerability, excluding hazard. Besides, they use the price in the exposure component, but they miss to consider the correlation between the building price and the building vulnerability.

 There is an inconsistency between sections,  in methods, they describe four building types and in the results, they describe seven. They divide the building into four-time stages which were not presented in methods and no justification was given to select these time stages, for example, 1910-1980 and 1980-1990 have the same construction material, but still they are separated.

The description of the reduction rate curve and the graph seems to be contradictory, a better description is required

 An explanation of why they used natural breaks to classify the vulnerability index is needed, what are the conceptual and practical meanings of these levels?

 In line 262 authors stated: The vulnerability distribution along the Huangpu River is obviously high, why is that obvious, are they still talking about vulnerability, or they are thinking more about risk?

 In line 302 What does it mean loss of exposure? Besides the figure 13 title said risk loss and the legend indicates flood risk, this needs to be corrected

 In line 313 authors talk about damage assessment, but no previous mention of that is given in the paper

 In line 314 authors mention that they introduce population and waterlogging data, but this needs a better explanation in the methods section. What kind of population data and waterlogging data were used? From what source? What is its spatial coverage? What is the period they cover?

 Other investigations are also using big data for flood risk that are not considered in the introduction and must be used to discuss their results.  

Reviewer 3 Report

This paper proposed a new research framework to assess urban vulnerability based on residential building big data. The framework can be widely applied to other cities to facilitate flood risk assessment and provide technical support for urban planning and risk management to enhance urban resilience. In the reviewer’s opinion, there are some major comments needed to be address. The main comments are listed below.

(1) Current introduction cannot clearly summarize the research status of this topic. The purpose and the novelty of this paper are not clearly defined. It would be better to include more works to discuss the state-of-the-art of the current issue and further refine the purposes and novelties of this paper in the Introduction.

(2) Section “Study area” should provide more information related to the current topic, such as climate, flood disasters, etc. This information can help this paper become more readable.

(3) Not all the building data of research region were collected for research. How about the representation of the current data? It would be better to provide an explanation in Section 2.2. In addition, what’s the usage of population data described in line 109~113?

(4) Section 2.3.1 “Asset and calculation of flood vulnerability”: What’s the meaning of “asset”?

(5) Section 2.3.2 proposed a new method to study vulnerability based on vulnerability curves. However, there are some questions that make me confused. How to build the vulnerability curve? Whether different buildings have different vulnerability curve? What is the physical meaning of the definite integral of the vulnerability curve? Please provide more details on the method description.

(6) As described in line 190, the Risk equals to the Exposure times Vulnerability. The unit of Exposure is yuan. And the unit of Vulnerability is m according to equation 1. Why the unit of risk is yuan instead of yuan*m? This makes me confused.

(7) As for the classification of building type, it is too fuzzy to understand. It would be better to give a detailed description.

(8) Some figures can be improved, such as Figure 4, 6, 7, 8, and 10. In addition, the locations of Zhejiang province and Jiangsu province in Figure 1 are wrong. Please verify all the figures and make sure the accuracy.

(9) The manuscript should be improved by a native English speaker.

In summary, there are many unclear points in the description of methods. It is hard for me to understand the results and discussion. This paper requires great effort to revise before accepting for publication.

Reviewer 4 Report

This manuscript, water-1871579-peer-review-v1- entitled "A method for assessing flood vulnerability based on big data of residential buildings -- a case study of Shanghai," is well written and has potential, but it should be more organized. This research investigates the taking Shanghai as an example, develops a new research framework to assess urban vulnerability based on residential building big data.

In my opinion, a careful revision of the English language should be carried out as there currently are some unclear sentences. The study seems to be well designed. The methodology and results are technically sound. Discussions on the scientific and practical values of the study, the limitations of proposed models, and future work are meaningful. I recommend accepting this manuscript after revision. The main concerns are as follows:

1)     The title section should be edited and rewritten since it should reflect the overall methodology.

2)     Quantitative results should be provided in the abstract to make it more comprehensive. Results of SVM and classification Should be added in the abstract section. Also, The main aim of the study should be clearly mentioned in the abstract.

3)     More recent references might support the first and second paragraphs of the introduction. Some references and literature are pretty old. There is no research reference for 2022. The authors should read and use the newly published papers in their research.

4)     More literature review about the other methods is needed. The manuscript could be substantially improved by relying and citing more on recent literature about contemporary real-life case studies of sustainability and/or uncertainty, such as the followings.

·       Vadiati, M., Rajabi Yami, Z., Eskandari, E., Nakhaei, M., & Kisi, O. (2022). Application of artificial intelligence models for prediction of groundwater level fluctuations: case study (Tehran-Karaj alluvial aquifer). Environmental Monitoring and Assessment194(9), 1-21.

·       Samani, S., Vadiati, M., Azizi, F., Zamani, E., & Kisi, O. (2022). Groundwater Level Simulation Using Soft Computing Methods with Emphasis on Major Meteorological Components. Water Resources Management, 1-21.

5)     For readers to quickly catch your contribution, it would be better to highlight significant difficulties and challenges and your original achievements to overcome them more straightforwardly in the abstract and introduction.

6)     Providing a comprehensive flowchart is highly recommended by researchers, so revise the flowchart, Fig.3, representing the overall methodology in the paper.

7)     Shanghai is adopted as the case study. What are other feasible alternatives? What are the advantages of adopting this case study over others in this case? How will this affect the results? The authors should provide more details on this.

8)     I suggest explaining more about Fig.s 12 and 13 since it is the core of the current research.

9)     Tab. 2, unfortunately, the authors did not try to discuss it in a specific way. A comprehensive discussion emphasizing would significantly improve the paper on the table.

10)  It is important to give a better description of the samples and the sampling protocol since we are trying to understand the data variability. What are the advantages of adopting these parameters over others in this case? How will this affect the results? More details should be furnished.

11)  Comparison of the current study with previous research, could be improved by more literature review.

12)  The limitations of the present study should be added to the paper, specifically for further research.

13)  It seems that conclusions are observations only, and the manuscript needs thorough checking for explanations given for results. The authors should interpret more precisely the results argument.

Round 2

Reviewer 1 Report

The author provided some information about missing points, but still, several items need to be addressed. See my comments below.

I suggest authors show exactly what has been changed in the manuscript. Several changes (font, color, etc.) can be seen, which may be a bit distracting for reviewers trying to focus on the main points. Some paragraphs seemed to be rewritten, but only slight changes can be seen upon closer examination. I recommend only highlighting the major changes.

The authors reported that the provided methodology can be applicable to big data analysis. I agree with this point. However, applicability doesn't mean that the current manuscript is a big data analysis. Technically, any method can be applied to a big data. So I suggest removing the big data part.

The authors provided more information about the vulnerability function, which I found insufficient to understand why the function is new. Why did the integral take place in the equation? Calculating some sort of volume didn't make sense. These functions are often created based on empirical datasets, which can sometimes be hard to find. However, why did the authors present the function as new when it is similar to what has been used in the literature? Please elaborate more.

The authors referred to Equation 2 for risk quantification. However, the equation alone doesn't give sufficient information on how the monetary risk is quantified. Exposure should be broken down to show how it is calculated.

I suggest revising number in Figure 12. (i.e. 13950 should be 13,950 for better readability) 

First floor height is a challenging measurement to acquire. The authors reported that they are working on obtaining the data. However, they should highlight this limitation in the manuscript.

Reviewer 2 Report

The authors adressed correctly the comments and suggestions. Now the article has more support, is easier to understand, and explains correctly the methods employed. 

Author Response

We would like to express our respect and gratitude to you for your valuable comments and suggestions on improving the quality of the paper.

Reviewer 3 Report

All of my comments have been addressed.

Author Response

(The authors gave the same response as above.)

Round 3

Reviewer 1 Report

Authors should explain more about the vulnerability function. Explanation of why integral took place should be stated.

Author Response

Comment: Authors should explain more about the vulnerability function. Explanation of why integral took place should be stated.

Accepted: Many thanks for the suggestion. A vulnerability curve reflects the ratios of potential damage that an in-risk element could suffer given different flood depths. Therefore, a vulnerability curve represents a series of values instead of a single score. It could not directly present the vulnerability level of an element, although the series of damage ratios would be typically higher for a vulnerable element than for a strong element. Thus, we use the integral of a vulnerability curve to represent the vulnerability level, which only rely on the characteristics of the exposed element, being independent on flood depths. This is a new attempt to evaluate flood vulnerability thanks to the curves. It could also overcome the drawbacks of index-based methods, which is sensitive to experience. We have added relevant explanation in Lines 161-165.